# Preparation and In Vivo Evaluation of a Lidocaine Self-Nanoemulsifying Ointment with Glycerol Monostearate for Local Delivery

**DOI:** 10.3390/pharmaceutics13091468

**Published:** 2021-09-14

**Authors:** Ji-Hyun Kang, Kwang-Hwi Yoo, Hyo-Young Park, Seung-Min Hyun, Sang-Duk Han, Dong-Wook Kim, Chun-Woong Park

**Affiliations:** 1College of Pharmacy, Chungbuk National University, Cheongju 28160, Korea; jhkanga@naver.com (J.-H.K.); dbrhkdgnl1@naver.com (K.-H.Y.); 2DONG-A Pharm. Co., Ltd., Yongin 17073, Korea; Hyogoo312@donga.co.kr (H.-Y.P.); hsm@donga.co.kr (S.-M.H.); tilldie@donga.co.kr (S.-D.H.); 3Department of Pharmaceutical Engineering, Cheongju University, Cheongju 28503, Korea

**Keywords:** hemorrhoids, local anesthetic ointment, lidocaine, glycerol monostearate, self-nanoemulsification

## Abstract

Lidocaine, a commonly used local anesthetic, has recently been developed into a number of ointment products to treat hemorrhoids. This study examined its efficient delivery to the dermis through the pharmaceutical improvement of hemorrhoid treatment ointments. We attempted to increase the amount of skin deposition of lidocaine by forming a nanoemulsion through the self-nanoemulsifying effect that occurs when glycerol monostearate (GMS) is saturated with water. Using Raman mapping, the depth of penetration of lidocaine was visualized and confirmed, and the local anesthetic effect was evaluated via an in vivo tail-flick test. Evaluation of the physicochemical properties confirmed that lidocaine was amorphous and evenly dispersed in the ointment. The in vitro dissolution test confirmed that the nanoemulsifying effect of GMS accelerated the release of the drug from the ointment. At a specific concentration of GMS, lidocaine penetrated deeper into the dermis; the in vitro permeation test showed similar results. When compared with reference product A in the tail-flick test, the L5 and L6 compounds containing GMS had a significantly higher anesthetic effect. Altogether, the self-nanoemulsifying effect of GMS accelerated the release of lidocaine from the ointment. The compound with 5% GMS, the lowest concentration that saturated the dermis, was deemed most appropriate.

## 1. Introduction

Hemorrhoid disease is the third leading outpatient gastrointestinal diagnosis, accounting for 3.3 million office and emergency department visits in the United States [1]. The self-reported incidence of hemorrhoids in the United States is 10 million per year, corresponding to 4.4% of the population [2,3,4]. In addition, although rarely serious, hemorrhoids often have a significantly negative impact on the quality of life [5]. Optimal pain control is required, particularly when patients are treated in an outpatient surgery setting. One important facet of the management of post-hemorrhoidectomy pain is the application of a multimodal pain control strategy [6,7,8]. The procedure-specific post-operative pain management (PROSPECT) working group has released Grade A recommendations for pain management after hemorrhoid surgery, including beyond-conventional oral analgesics, oral diosmin, metronidazole, laxatives, local application of lidocaine, glyceryl trinitrate, cholestyramine ointments, and infiltration with long-acting local anesthetics [9,10].

Lidocaine is a widely used local anesthetic that relieves pain, burning, and itching caused by hemorrhoids [11,12]. Specifically, lidocaine produces analgesia by blocking voltage-gated sodium channels, which are responsible for the propagation of action potentials [7,13]. This molecule is characterized by the onset of its surface anesthetic action within a few minutes, providing rapid relief from pain and itching [13]. When applied locally, lidocaine needs to permeate through the skin to act as an anesthetic or analgesic. This outer layer of skin is made up of keratinized, stratified squamous epithelial cells that form a permeability barrier that prevents the movement of water [14]. Recently, lidocaine has been applied to various topical delivery dosage forms, such as microneedles and depots [15,16]. However, these dosage forms are difficult to use for hemorrhoids. It has been proposed that local dosage compounds take up water from sweat or trans-epidermal water loss, then form a self-emulsion.

Local delivery can be defined as the application of a drug-containing formulation to the skin in order to directly treat cutaneous disorders, such as acne, or the cutaneous manifestations of a general disease, such as hemorrhoids [17]. Local drug delivery offers advantages such as ease of delivery, a cooperative patient, increased compliance, and avoidance of first-pass metabolism [15,18]. Local anesthetics have always been of considerable clinical benefit in reducing cutaneous pain associated with various medical procedures. The stratum corneum layer of the epidermis is the main hurdle, and penetration through it is the rate-limiting step in the percutaneous absorption of local anesthetic drugs. As the sensory nerve endings responsible for pain sensation are mainly present in the dermis, local anesthetics should aim to penetrate this layer during local administration [19]. New drug delivery technologies and penetration enhancers may help avoid some of these issues [20]. An alternative delivery strategy used to increase the penetration of local anesthetics is the incorporation of the drug into innovative colloidal carrier delivery systems, such as microemulsions [21]. An emulsion is a system that contains water and/or oil nanodomains coexisting in thermodynamic equilibrium, owing to the presence of a surfactant film at the oil/water interface [22].

The use of fatty acids in local treatments enhances skin permeability, mainly due to disturbance of the stratum corneum [23,24,25]. More specifically, fatty acids have been found to increase the fluidic movement of surface lipid structures, thereby decreasing the phase transition temperatures of lipid structure [26,27]. Fatty acids are also used as emulsifiers in pharmaceutical and cosmetic products [28]. Lipid glycerol monostearate (GMS) is composed of a single fatty acid chain attached to a glycerol backbone [29]. GMS molecules self-assemble in water and oil into several types of mesophases, and the polymorphic and mesomorphic properties of GMS can affect the behavior of emulsion systems [30,31]. The lamellar structure of GMS is similar to the structure of the human stratum corneum; therefore, the hydrated lamellar phase could significantly increase the water content of the stratum corneum [32,33]. There is much evidence that hydration induces a significant increase in drug permeability [34,35,36]. The concept of self-emulsifying drug delivery systems (SEDDSs) was pioneered in the 1960s, when components of poor aqueous solubility were incorporated into mixtures of lipophilic and hydrophilic excipients to enhance the solubility of lipophilic substances [37]. Recently, research has been conducted on applying SEDDS to the skin [38,39,40]. It has been proposed that local dosage compounds take up water from sweat or trans-epidermal water loss, then form a self-emulsion [41].

This study aimed to improve lidocaine ointments to relieve hemorrhoid pain using a petrolatum base [42,43]. Although the petrolatum is effective in relieving itching from hemorrhoids, it may not be ideal as a carrier, due to slow drug release [44]. To overcome the disadvantages of petrolatum base ointment, we prepared the ointment through self-nanoemulsion with GMS, in order to improve its efficacy. After producing ointments with different proportions of GMS, the physicochemical properties were evaluated, and the effect of GMS on permeability was studied through ex vivo permeability tests. In addition, we visualized the improved penetration through ex vivo experiments using Raman mapping. Finally, we performed an in vivo tail-flick test to confirm the local anesthetic effect of lidocaine ointments.

## 2. Materials and Methods

### 2.1. Materials

The lidocaine base was a gift from DONG-A Pharm Co., Ltd. (Seoul, Korea). GMS was obtained from Gattefosse Corporation (Lyon, France). Petrolatum was purchased from Wonpoong Pharm Co., Ltd. (Seoul, Korea). Medium-chain triglyceride oil (MCT oil) was obtained from KLK OLEO (Klang, Selangor, Malaysia). Other excipients were supplied by DONG-A Pharm Co., Ltd. (Seoul, Korea). Reference Product A (Ref. A), Prepain^®^ ointment, was obtained from Ildong Pharm. Co., Ltd. (Seoul, Korea). Water was purified by filtration in the laboratory. High-performance liquid chromatography (HPLC)-grade solvents were used for analysis. HPLC-grade ethanol and acetonitrile were purchased from Honeywell Burdick & Jackson (Muskegon, MI, USA). Sprague Dawley^®^ rats (SD rats) and Institute of Cancer Research mice (ICR mice) were purchased from Samtako Corporation (Osan, Korea).

### 2.2. Methods

#### 2.2.1. Preparation and Rheological Characterization of Lidocaine Ointment 

The ointments containing lidocaine were prepared using a fusion method [45,46]. Each oil excipient, including petrolatum, MCT oil, GMS, and vitamin E-acetate, was first melted in a water bath at 70 °C. Active pharmaceutical ingredients such as lidocaine, allantoin, prednisolone acetate, and dl-methylephedrine hydrochloride were then gradually added to the molten material and then mixed at 6000 rpm and 70 °C for 5 min using a homogenizer (Homogenizing mixer Mark II model 2.5, Primix Corp., Tokyo, Japan). The mixture was cooled to 20 °C under constant stirring. 

The composition of each ointment is listed in Table 1. L1, L2, and L3 are petrolatum-based ointments containing 0%, 5%, and 10% of GMS, respectively. The mixtures were in vitro research formulations containing lidocaine alone as an active pharmaceutical ingredient. Formulations L4 (Hemo ointment, Dong-A Pharm., Co., Ltd., Seoul, Korea), L5 (ChioMAX ointment, Dong-A Pharm., Co., Ltd., Seoul, Korea), and L6 were prepared for in vivo testing.

The viscosity test was performed in an RV-7 viscometer (DV-III Ultra RV, AMETEK Brookfield Inc., Middleboro, MA, USA) at 25 °C and 50 rpm for 3 min. The minimum extrusion force was determined with a TA-XTplusC texture analyzer (Stable Micro Systems Ltd., Surrey, Godalming, UK). After the tube was completely filled with the ointment, the minimum force at which the ointment started to flow when compressed was measured. Each experiment was conducted in triplicate.

The pH measurement was conducted by pH meter (Sevencompact, Mettler Toledo Inc., Greifensee, Switzerland) with a 10-fold diluted ointment using distilled water.

#### 2.2.2. Physicochemical Characterization of Lidocaine Ointment

To determine the droplet size and polydispersity index the Portal ELSZ Zeta-size analyzer (Otsuka Electronics, Osaka, Japan) was employed. The measurements were made at 90° at room temperature (25 °C). One milliliter of the sample was placed in the cuvettes and the lidocaine ointments were diluted (2-fold in phosphate buffer solution (PBS, pH 7.4). Each experiment was conducted in triplicate [18].

The thermal properties of the lidocaine, other excipients, and prepared lidocaine ointments were analyzed using a DSC Q2000 (TA Instruments, New Castle, DE, USA) thermal analyzer system. The samples were weighed, loaded into an aluminum pan, and analyzed at a heating rate of 10 °C/min over a temperature range of 40 °C to 120 °C. The thermal responses of the prepared samples were calculated using the TA Advantage/Universal Analysis software (v5.2.6, TA Instruments, New Castle, DE, USA).

X-ray diffraction (XRD) patterns of lidocaine, other excipients, and lidocaine ointments were analyzed using a D8 Discover with GADDS (Bruker AXS, Karlsruhe, Germany) at a wavelength of 1.54 Å. The 2θ scans were conducted between 5° and 60°.

Raman spectroscopic measurements were performed using a RAMANtouch device (Nanophoton Corp., Osaka, Japan) equipped with a CCD camera and a diode laser. The spectra of the individual components and formulations were acquired by 10 scans for a 10 s exposure time. Measurements were carried out with a 532 nm laser with a power level of approximately 1 mW at a slit width of 50 μm in the point measurement mode. Lidocaine, petrolatum, MCT oil, GMS, and prepared ointments were analyzed. In addition, raw SD rat skin and ointment-treated SD rat skin were analyzed. Ointment-treated SD rat skin was treated with ointment for 6 h, and the remaining ointment was removed using an alcohol swab, then analysis was performed. 

The morphology and structure of the diluted ointment were characterized using transmission electron microscopy (TEM). First, the diluted ointment was loaded onto a copper grid coating (TED PELLA Corporation, Redding, CA, USA) by soaking for 1 min. Next, the samples were dyed using 1% (*w*/*v*) phosphotungstic acid solution for 10 s. The sample was dried for approximately 3 h at 25 °C. Images were obtained with an accelerating voltage of 200 kV using a LIBRA 120 instrument (Carl Zeiss, Oberkochen, Germany) [18].

#### 2.2.3. In Vitro Dissolution Test of Lidocaine Ointment

The in vitro dissolution behavior of the lidocaine ointments was evaluated using a dialysis membrane (Slide-A-Lyzer^®^ mini dialysis devices 10 K MWCO, ThermoFisher Scientific Inc., Waltham, MA, USA). Precisely, 1 g of ointment was added to the device, and the device was placed into a 50 mL conical tube containing the dissolution medium, 45 mL of PBS. The conical tube was gently shaken at 100 rpm and maintained at 37 ± 1 °C in a water bath (BS-06, Jeiotech^®^, Daejeon, Korea). At a defined time, 200 μL of the medium was added, and the same volume of fresh PBS was added. Drug content was quantified using HPLC. The HPLC system (Ultimate 3000 series HPLC system, Thermo Scientific, Waltham, MA, USA) was operated at 265 nm with an L1 packing 250 mm × 4.60 mm, 4 μm column (Phenomenex, Torrance, CA, USA). The mobile phase, consisting of acetonitrile and buffer (5% glacial acetic acid in distilled water, adjusted to pH 3.4 NaOH) at a 20:80 (*v*/*v*) ratio, was eluted at a flow rate of 1.0 mL/min. The column temperature was maintained at 30 °C, and the volume of each injected sample was 20 μL.

#### 2.2.4. Ex Vivo Skin Permeation and Raman Mapping of Lidocaine Ointment

The ex vivo skin deposition test and Raman mapping of lidocaine ointment (L1, L2, and L3) were conducted using SD rat skin. Tests of L4, L5, L6, and Ref. A was performed using the human cadaver skin. The skin was hydrated with PBS and mounted in a Franz diffusion cell. PBS (pH 7.4) containing ethanol (50% *v*/*v*) was used as the receptor phase. The prepared lidocaine ointments and the Ref. A (3% *w*/*w* lidocaine) was applied to the donor compartment (approximately 2.0 cm^2^). An ex vivo skin test (*n* = 4) was performed over a 6 h period, after which the excess ointment was removed with an alcohol swab. Adhesive tape (3M Scotch^®^, Paul, MN, USA) was used to separate the stratum corneum. The epidermis and dermis were cut into six pieces [18]. The pieces were then soaked in 5 mL of the mobile phase by sonication. The extracted sample was filtered using a 0.45 µm pore 25 mm GD/X syringe filter (Whatman, Maidstone, UK). The filtered sample was diluted 2-fold with the mobile phase. The lidocaine concentration in each sample was quantified using HPLC.

Raman mapping was performed to determine the distribution of lidocaine throughout the skin depth. After the ex vivo skin permeation test for 6 h (*n* = 3), the skin was cut into 50 µm slices from top to bottom using a cryomicrotome (Microm HM450, Thermo Scientific, Waltham, MA, USA). Up to approximately 300 µm of the skin was obtained and cut vertically to determine the difference in lidocaine by depth. The skin slice was placed on a glass slide (Marinefield Superior, Lauda-Königshofen, Germany) and examined by Raman mapping (RAMANtouch, Nanophoton Corp., Osaka, Japan). In each experiment, the laser light (532 nm, 50 mW) was focused on the sample. The magnification of the sample was 20. The acquisition time was 0.1 sec and the slit size was 50 µm. The point-by-point mode was used for mapping. The analysis was performed at 1670 cm^−1^. The view size was a square with 100 µm × 100 µm. The pixel size was 1.0 µm × 1.0 µm. In the Raman mapping, a high concentration of lidocaine is presented as red color, whereas a relatively low concentration of lidocaine was displayed as blue color.

#### 2.2.5. In Vivo Tail-Flick Test of Lidocaine Ointment

The Chungbuk National University Institutional Animal Care and Use Committee approved the experimental protocols and animal care methods used in this study. In the present study, 20 male ICR mice were divided into five groups (*n* = 4). The mice were housed at a temperature of 20 °C and relative humidity of 40% during the experiment. The mice were acclimatized for 7 days before the tail-flick test. To perform the test, a 1.5 cm portion of the tail was heated by a radiant heat source. Tail flick latency (TFL) time was measured by turning on the heat source to the flicking of the tail. The mean time of four measurements before the administration of ointment was performed as the baseline latency. Mice were selected for testing with a mean baseline latency time of 2–4 s. A 10 s cutoff time was applied to avoid thermal damage to the tail at all points in the experiment [47]. 

The five groups of mice were treated with either 50 mg of L4 ointment (L4 group), 50 mg of L5 ointment (L5 group), 50 mg of L6 ointment (L6 group), 50 mg of Ref. A (Ref. A group), or normal saline (normal saline group). The time points of the tail-flick test were 0, 30, 50, 60, 75, 90, and 105 min after drug administration. The area under the receiver operating characteristic curve (AUC) was calculated from the TFL-time profile of each formulation using the linear trapezoidal rule. 

#### 2.2.6. Statistical Analysis 

Statistically significant differences were evaluated using one-way analysis of variance (ANOVA) with the Least Significant Difference (LSD) post hoc test using SPSS version 23 (SPSS Inc., Chicago, IL, USA). Statistical significance was set at *p* < 0.05.

## 3. Results

### 3.1. Physicochemical Characterization of Lidocaine Ointment

Physicochemical characterization of lidocaine ointment was performed. The droplet size and polydispersity index (PDI) of the diluted lidocaine ointment were determined (Table 2). The examination of L1 did not form a homogeneous emulsion. When L2 was diluted to 1:2, the droplet size was 117 nm, and the PDI was 0.28, indicating the formation of droplets of uniform size. L3 also had a droplet size at a 1:2 dilution of 108 nm and a PDI of 0.30, indicating a uniform droplet.

DSC thermograms of the raw materials and lidocaine ointment are shown in Figure 1. When petrolatum was examined with DSC analysis up to 120 °C, melting occurred with a small and broad peak at 45 °C. The thermogram of GMS and lidocaine showed a sharp endothermic peak at 65 °C and 70 °C, which was indicative of the melting and crystalline nature of GMS and lidocaine. This is similar to the previously reported melting temperature (Tm) values of GMS and lidocaine [48]. The thermogram of the L2 and L3 ointment, made with 5% and 10% GMS, showed a widened and split endothermic peak from 50 °C to 60 °C. The higher the proportion of GMS, the higher is the Tm. However, this difference was not statistically significant. 

The correlation of XRD data (Figure 2) and the DSC results was good. The crystal form of raw lidocaine showed sharp and narrow peaks whereas, petrolatum, MCT oil, and GMS diffraction patterns indicate amorphous phases which broad peaks at 2θ = 19°, 21°, and 23°, respectively.

Raman spectroscopy analysis can provide further confirmation of the stability and crystallinity of the prepared lidocaine ointments. In this study, raw lidocaine, petrolatum, MCT oil, GMS, and prepared ointments were analyzed. The spectrum of raw lidocaine was similar to that of the reference, showing absorption peaks at 3044, 2945, and 1670 cm^−1^ (Figure 3A) [49]. 

The shape of the droplet of the diluted ointment was confirmed by TEM (Figure 4). The magnification of the TEM image is ×20,000, and each scale bar indicates 400 nm. L1 does not form a droplet, and its size is not uniform. On the other hand, L2 and L3 formed a uniform droplet with a size of approximately 100 nm. 

### 3.2. In Vitro Dissolution Test of Lidocaine Ointment

The in vitro dissolution behavior of the lidocaine ointments was evaluated using a dialysis membrane (Slide-A-Lyzer^®^ mini dialysis devices 10 K MWCO, ThermoFisher Scientific Inc., Waltham, MA, USA). Figure 5A shows the dissolution profile of lidocaine ointment. L2 and L3 with GMS exhibited approximately twice as much dissolution as L1 at 1 h. The amount of lidocaine released 6 h after L1 was 695.7 μg/cm^2^, while L2 released 815.1 μg/cm^2^ and the L3 ointment released 1021.6 μg/cm^2^. Additionally, the kinetics of the release data showed the best fit with the zero-order release model (Figure 5B). The release kinetic constants were 114.2 for L1, 132.0 for the L2, and 162.5 for the L3. The higher the proportion of GMS, the higher the release kinetic constant; there is a very high correlation (0.97) between the proportion and the release constant. 

### 3.3. Ex Vivo Skin Permeation Test and Raman Mapping of Lidocaine Ointment

In this experiment, the depth of lidocaine penetration was determined by Raman mapping, for each formulation. The 1670 cm^−1^ peak, which is a specific Raman absorption band of lidocaine, was monitored. The highest concentration of lidocaine was observed in the stratum corneum shown, while deeper skin samples presented the lower lidocaine concentration (Figure 6). At L1 treated animals, lidocaine was not detected in 200 μm depth skin layers. In contrast, in the case of L2 and L3 ointments, lidocaine was detected up to 300 μm depth layers. Generally, the thickness of the epidermis is approximately 100 μm [50]; therefore, L1 penetrates to the upper end of the dermis, while L2 and L3 penetrate deeper into the dermis.

These Raman mapping results are very similar to those of the in vitro permeation test performed using SD rat skin (Figure 7). For L1, 3.3 mg/cm^3^ of lidocaine was deposited on the stratum corneum at the 6 h mark and 8.0 mg/cm^3^ of lidocaine penetrated the dermis; L2 had 5.3 mg/cm^3^ and 19.7 mg/cm^3^; L3 had 10.1 mg/cm^3^ and 21.7 mg/cm^3^. 

In vitro permeation tests were performed on human cadaver skin using the compositions L4, L5, L6, and Ref. A (Figure 8). L5 and L6 showed a permeation profile close to zero-order with almost no lag time (Figure 8A). Ref A., L4, L5, and L6 permeated by 21.2, 19.6, 33.5, and 41.7 μg/cm^2^ in the receptor phase at 6 h, respectively (Figure 8B). L5 delivered significantly greater amounts of lidocaine than Ref A and L4, while L6 delivered significantly greater amounts than Ref A, L4, and L5 (*p* < 0.005). The correlation between the flux and GMS ratio was very high, with a value of R2 0.999. 

### 3.4. In Vivo Tail-Flick Test

Figure 9 and Figure 10 show the latency profile and AUC of the tail-flick test. AUCs of normal saline, Ref. A, L4, L5, and L6 groups were 439.8, 512.3, 583.0, 706.1, and 778.9 s·min, respectively (Table 3). All groups had a statistically higher AUC than the normal saline group. In addition, the AUC was significantly higher than that in Ref. A in all three groups (L4, L5, and L6). In particular, L5 and L6 were more significant than L4, with *p*-values < 0.005. There was no statistically significant difference in the AUC between L5 and L6. The effect folds were calculated for a relative comparison of anesthesia effects [51]. Comparing effect folds using AUC, L5 was 1.38 times more effective than Ref. A, while L6 was 1.52 times more effective. This result is very similar to the amount of lidocaine deposited in the dermis (Figure 7). 

## 4. Discussion

As shown in Table 2, smaller droplets increased the skin permeation rate. In the L2 and L3 formulations (containing GMS), nanoemulsions with uniform nanosized droplets were formed. In that case, MCT oil forms an oil core, while GMS acts as a surfactant surrounding it to form nanosized droplets [52]. Unlike L1, such formulations rapidly dissolved and released the GMS and lidocaine dispersed in the petrolatum carrier by forming droplets at the interface with a small amount of water derived from the skin. In this system, droplets are formed at the interface with a small amount of water derived from the skin, which infiltrates the skin immediately after droplet formation so that the pharmaceutical is released rapidly from the ointment. Many fatty acids enhance the droplet as a negative charge through the disruption and fluidization of the stratum corneum layer so that the droplet can rapidly penetrate the skin [23,24,25]. Therefore, the droplets formed by GMS can disrupt the stratum corneum and quickly infiltrate the skin as a nanoemulsion.

The thermograms of the lidocaine ointments were most affected by the presence of petrolatum and GMS (Figure 1). The thermograms of the prepared L1 ointment made without GMS were similar to those of the raw petrolatum. GMS undergoes polymorphic transformation by heating. This indicates that GMS forms a nanoemulsion structure during the ointment preparation process [31]. In addition, none of the lidocaine ointments exhibited characteristic raw lidocaine endothermic melting peaks, indicating that lidocaine exists in an amorphous state within the lidocaine ointments. 

The sharp and narrow peaks of lidocaine disappeared in the XRD patterns of L1, L2, and L3 (Figure 2). Based on these results, lidocaine was dissolved in an ointment in an amorphous form. The amorphous form generally has higher solubility and bioavailability than those of the crystalline form [53,54]. Therefore, the amorphous form is expected to improve bioavailability by enhancing skin permeability.

In Figure 3A, the 3044, 2945, and 1670 cm^−1^ peaks correspond to methyl antisymmetric stretching, CH2 antisymmetric and symmetric stretching, and the C=O bond of the amide bond [55]. The peaks of raw lidocaine disappeared in the ointment. This is because lidocaine is completely dissolved and its crystallinity is lost [56,57]. These results agree with those of a previous study on local dosage forms of lidocaine [50]. Next, Raman spectra analysis of raw SD-rat skin and ointment-treated SD-rat skin was also performed. The peaks at 1670 cm^−1^ did not overlap with other excipients and were not found in raw SD-rat skin, but were found in ointment-treated SD-rat skin. Excipient-derived peaks were also confirmed in ointment-treated SD rat skin, as they were derived from the excipients merged into the skin. In addition, 1670 cm^−1^ peaks, which were not visible in the ointment, were identified in ointment-treated SD rat skin. This is because the intensity of the 1670 cm^−1^ peak increased even after removal of the ointment, as lidocaine migrated from the ointment to the skin. Since the excipient did not migrate to the skin, the intensity of the excipient-derived peak decreased after the ointment was removed. Therefore, the intensity of the peak at 1670 cm^−1^ was relatively higher in ointment-treated SD rat skin. This wavelength was used for Raman mapping and subsequent experiments (Figure 3B).

The TEM image in Figure 4 clearly confirms the presence of self-nanoemulsification. L1 is present in merged, non-uniform, large-sized oil masses that do not form clear droplets. On the other hand, L2 and L3 displayed round and homogenous shading with a droplet size of approximately 100 nm. This clearly supports the results for droplet size using a zeta-size analyzer. Owing to the presence of GMS, the oil phases of the L2 and L3 ointments were able to achieve a self-nanoemulsion.

As the dissolution progressed, L2 slowed down, while L1 and L3 maintained their dissolution rates (Figure 5A). This is because the GMS contacted the water of the hydrated membrane to form a self-nanoemulsion and penetrate the membrane. By forming a nanoemulsion at the interface between the hydrated membrane and the ointment, the droplet rapidly permeated and then dissolved, in order to maintain the sink state, thereby allowing the rapid dissolution of lidocaine and GMS from the ointment to the interface. L1 is released according to zero-order because it does not have GMS and is released by simple diffusion along the concentration gradient of lidocaine. In addition, L3 is comparatively rich in GMS, so droplets are steadily formed and released at zero-order faster than L1. On the other hand, it is possible that L2 does not have sufficient GMS, and the dissolution rate decreases in the latter half. However, when actually applied to the skin, unlike the dissolution environment, only a very small amount of water is present; therefore, GMS deficiency does not occur. In Figure 5B, high release kinetic constants indicate that the system by which GMS forms droplets allows the drug to be released rapidly from the petrolatum, which is a highly lipophilic ointment agent, and allows the drug to penetrate the skin. This means that it is faster and has a higher residual amount in the skin and can effectively deliver the drug to the dermis.

As shown in the Raman mapping results (Figure 6), L1 was relatively less permeated than L2 and L3. Since it does not penetrate deep into the dermis, it is likely that a significantly lower amount was delivered. In addition, L3 had a larger amount of deposit in the stratum corneum than L2 because the self-nanoemulsion progressed more rapidly due to the large amount of GMS, and lidocaine was dissolved from the ointment faster. L2 and L3, with GMS, show not only a larger amount of lidocaine penetration, but also a deeper deposition than L1 without GMS, so it is expected that the onset to reach the anesthetic effect is fast and the anesthetic effect is higher. The presence or absence of self-nanoemulsifying changes depends on the presence or absence of GMS. Therefore, L1 was unable to self-nanoemulsify, so the rate of dissolution from the ointment to the skin was slow and the rate of penetration into the deep skin was slow (Figure 7). On the other hand, the GMS present in L2 and L3 ensured a high driving force and high penetration owing to the high dissolution rate and self-nanoemulsifying effect. Raman mapping by skin depth and the permeation test with SD rats did not show a significant difference depending on the amount of GMS, only its presence. 

The permeation tests using human cadaver skin also showed that the influence of the GMS ratio was the largest (Figure 8A). Similar to the previous results, the presence or absence of GMS had a great influence on how quickly the drug was released from the petrolatum base ointment. Since the dissolution of lidocaine from the ointment was accelerated by the self-nanoemulsifying effect of GMS, there was no lag time for both L5 and L6, showing a permeation profile close to zero order. As a result, as the ratio of GMS increased, the flux also increased, and linearity was possible (Figure 8B). In conclusion, through the self-nanoemulsifying effect of GMS, L5 and L6 showed higher permeability than Ref. A and L4.

In all formulations, latency tended to increase up to approximately 60 min overall and then decreased. However, L5 and L6 maintained a high latency of up to 105 min, with a relatively small decrease (Figure 9). As shown in Figure 10, the latency was significantly higher in L4, L5, and L6 than in the normal saline group at 30 min (Figure 10A).

However, there was no difference between L4 and Ref. A at 90 min. L5 and L6 showed higher latencies than those in Ref. A at all-time points (Figure 10B,C). The self-nanoemulsifying effect of GMS allowed lidocaine to penetrate the skin from the ointment and rapidly reach the intradermal concentration, showing an anesthetic effect; as GMS exists at a higher concentration, the latency of the L5 and L6 groups is higher in the tail-flick test. Finally, the correlation between the AUC and GMS ratios was also high (R2 = 0.96), but was relatively low compared to the other parameters (Figure 10D). This is because the AUC tended to deviate slightly from the linearity at L5. These results suggest that the difference between GMS 5% and GMS 10% had no significant effect in vivo. This result is similar to the fact that the amount deposited in Raman mapping (Figure 6) and dermis (Figure 7) did not differ significantly between the two formulations. After being rapidly released from the ointment by the GMS, lidocaine is transferred to the dermis. It is possible that the amount deposited in the dermis and transferred to the receptor phase was not significantly different from the amount originally deposited in the dermis. Lidocaine was rapidly released from the ointment base by GMS in vivo. It is expected that it was transferred to the dermis over time, and when the amount exceeded a certain level, it was transferred to the blood. Therefore, there would have been no difference between L5 and L6 in the skin, similar to the deposited amount and Raman mapping results for the dermis evaluated in vitro.

## 5. Conclusions

We conducted a pharmaceutical study to take advantage of the self-nanoemulsifying effect of GMS, allowing lidocaine to be released rapidly from the petrolatum ointment and rapidly penetrate the dermis. The self-nanoemulsifying effect of GMS was confirmed through an in vitro dissolution test, which resulted in rapid release from the ointment. Due to these properties, in vitro skin permeation also shows a rapid transition to the receptor phase with GMS, while a larger amount of lidocaine was deposited in the dermis. In the in vitro permeation test, the penetrated amount in the receptor phase and flux increased as the amount of GMS increased, and the profile showed a zero-order form with no lag time, which had a linear correlation with the ratio of GMS. However, there was no difference in the amount deposited in the dermis, as confirmed by Raman mapping. In the case of L1 without GMS, the drug permeated to a depth of 200 μm, which is the position that seems to be the top of the dermis, while in L2 and L3, which contained GMS, the drug was distributed to a deeper layer of the dermis. After saturation of the dermis above a certain amount, the drug shifts to the receptor phase, so it is considered that there is no significant difference in the amount deposited in the dermis between L2 and L3, beyond that which was necessary for saturation. Furthermore, similar results were obtained from the in vivo tail-flick test. In the case of the Ref. A and L4 ointment without GMS, the latency between stimulus and tail-flick was higher than that in the normal saline group, but there was no significant difference between Ref. A and L4. However, L5 and L6 had significantly higher latency and AUC, and were more effective than Ref. A, L5 at 1.38 times, and L6 at 1.52 times. Additionally, improvement of skin deposition was confirmed through in vitro Raman mapping, and the superiority of local anesthetic effect was confirmed through an in vivo tail-flick study. In conclusion, the self-nanoemulsifying effect of GMS was applied to improve local anesthetic ointment, primarily for the treatment of hemorrhoids. 

## Figures and Tables

**Figure 1 pharmaceutics-13-01468-f001:**
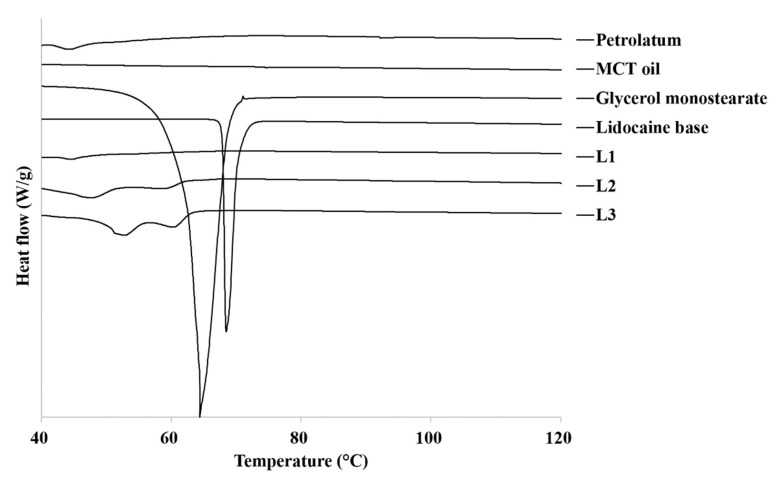
DSC thermograms of raw materials and prepared lidocaine ointments. MCT, medium-chain triglyceride.

**Figure 2 pharmaceutics-13-01468-f002:**
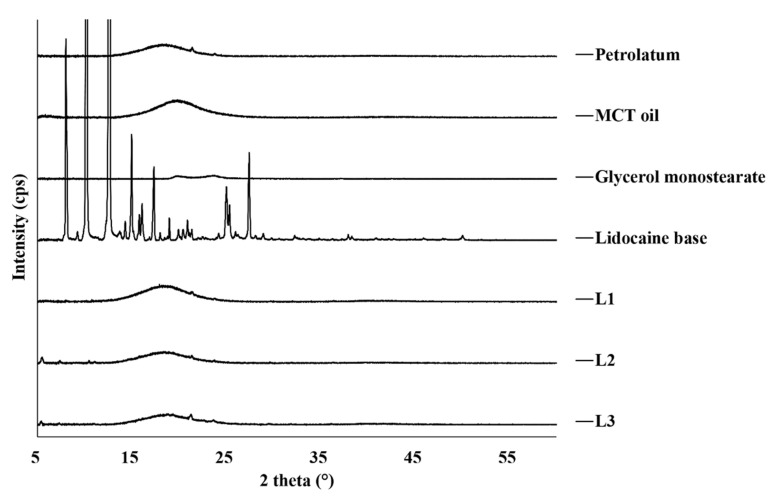
X-ray diffraction patterns of the raw materials and prepared lidocaine ointments. MCT, medium-chain triglyceride.

**Figure 3 pharmaceutics-13-01468-f003:**
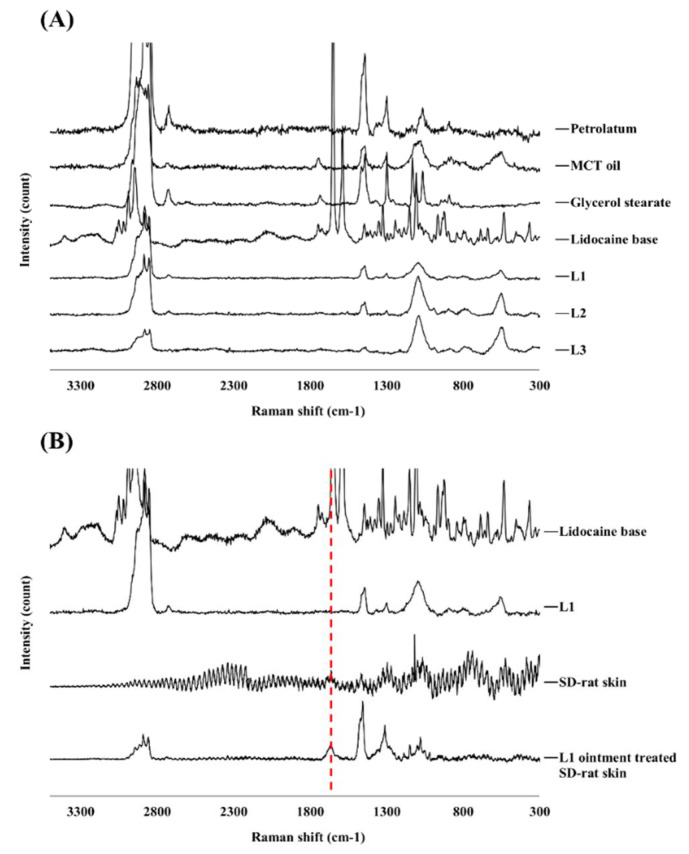
Examination of compounds by Raman spectra. (**A**) Raman spectra of the raw materials, and prepared lidocaine ointments; (**B**) Raman spectra of lidocaine base, L1 ointment, SD-rat skin, and L1 ointment treated SD-rat skin (red line: Raman peak of lidocaine base at 1670 cm^−1^). SD: Sprague-Dawley.

**Figure 4 pharmaceutics-13-01468-f004:**
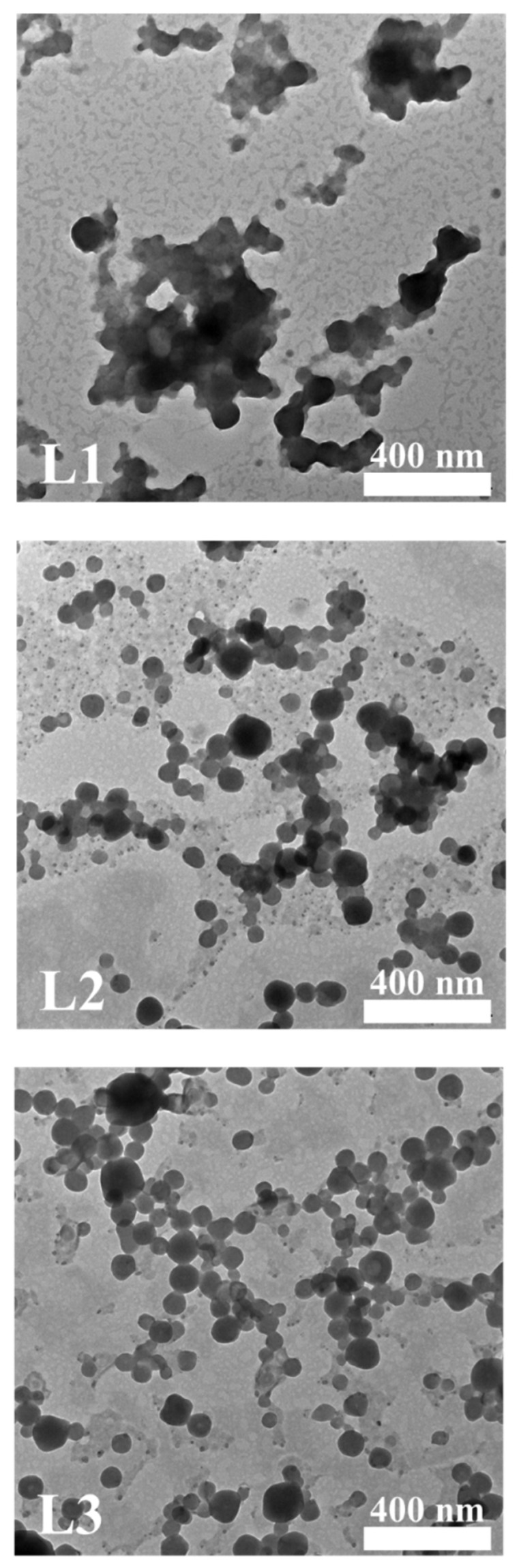
TEM images of diluted lidocaine ointments (magnification: ×20,000, scale bar: 400 nm). TEM: transmission electron microscope.

**Figure 5 pharmaceutics-13-01468-f005:**
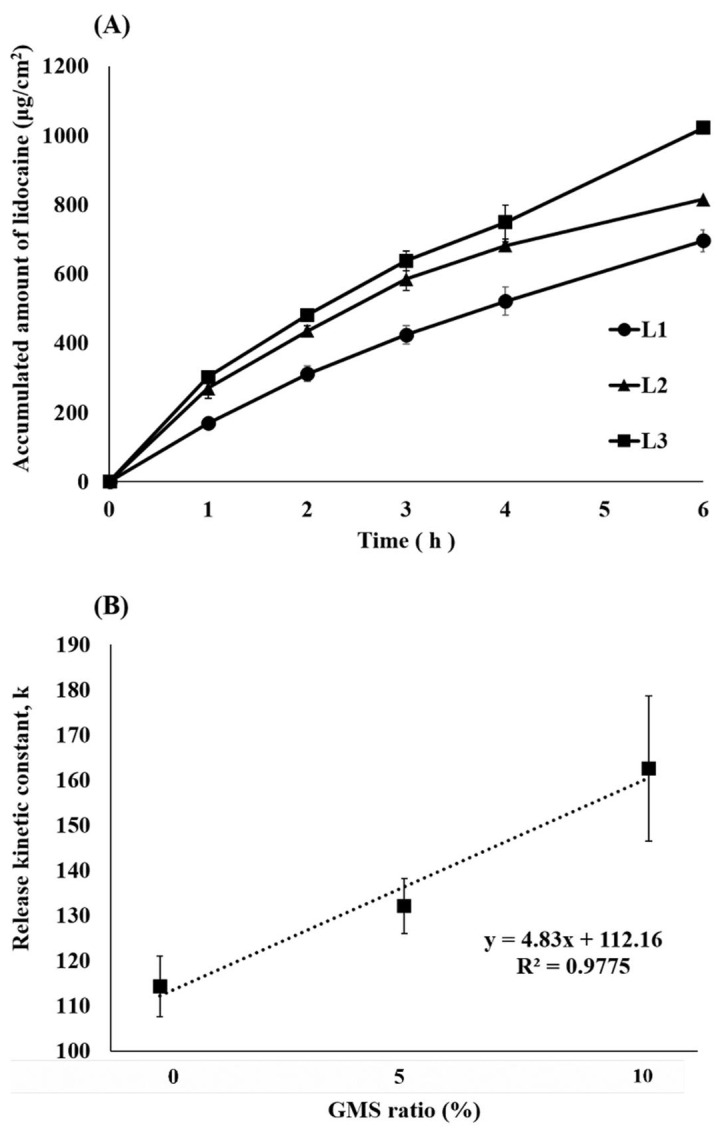
In vitro dissolution tests of lidocaine ointment. (**A**) Dissolution profile of prepared lidocaine ointments; (**B**) Correlation between kinetic constant (k) and GMS ratio (%) (mean ± standard deviation, *n* = 4). GMS: glycerol monostearate.

**Figure 6 pharmaceutics-13-01468-f006:**
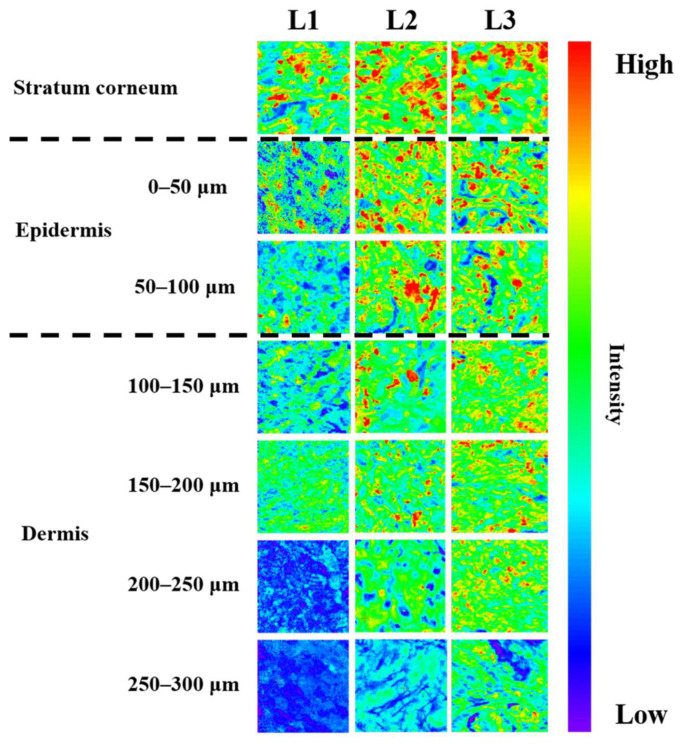
Raman mapping of the stratum corneum and other skin layers at different depths after ex vivo skin permeation test using lidocaine ointment.

**Figure 7 pharmaceutics-13-01468-f007:**
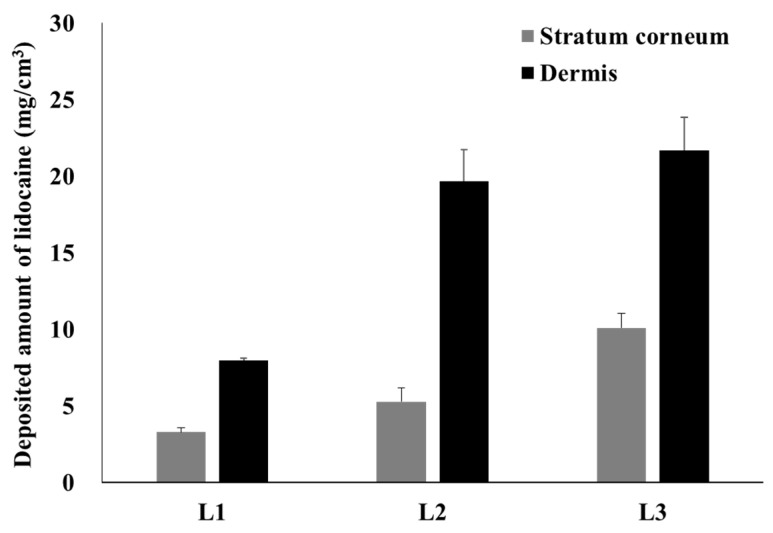
Deposited amount of lidocaine in the stratum corneum and other skin layers with SD-rat skin (mean ± standard error, *n* = 4). SD: Sprague-Dawley.

**Figure 8 pharmaceutics-13-01468-f008:**
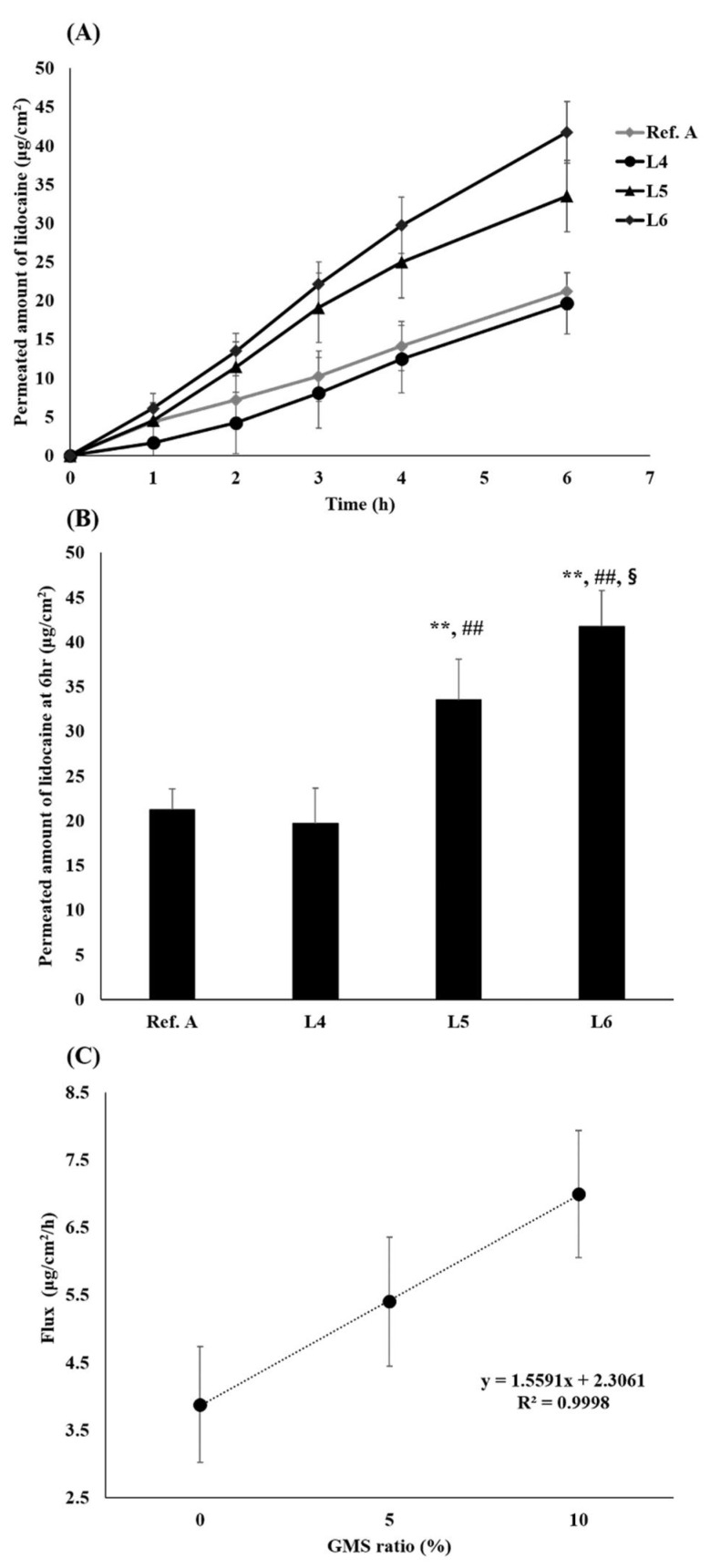
Permeation tests of L4, L5, L6 ointments. (**A**) Permeation profile of prepared lidocaine ointments and reference product A with cadaver skin; (**B**) Permeated amount of lidocaine at 6 h; (**C**) Correlation between flux and GMS ratio (%) (mean ± standard deviation, *n* = 4). GMS: glycerol monostearate. § ANOVA, LSD, *p*-value < 0.05 compared with L5 group. ** ANOVA, LSD, *p*-value < 0.005 compared with Ref. A group. ## ANOVA, LSD, *p*-value < 0.005 compared with L4 group.

**Figure 9 pharmaceutics-13-01468-f009:**
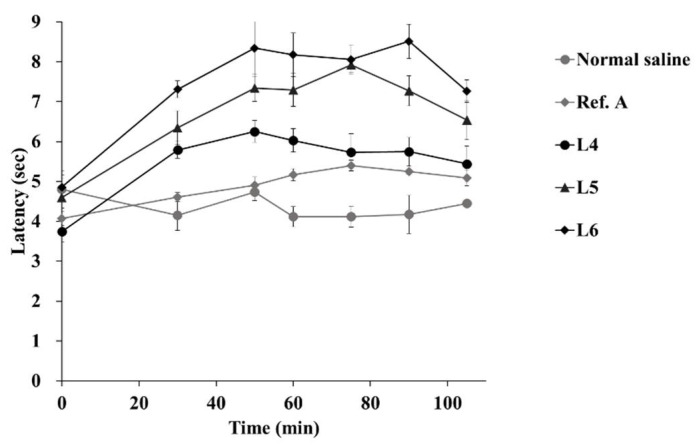
Latency profile of in vivo tail-flick test (mean ± standard error, *n* = 4).

**Figure 10 pharmaceutics-13-01468-f010:**
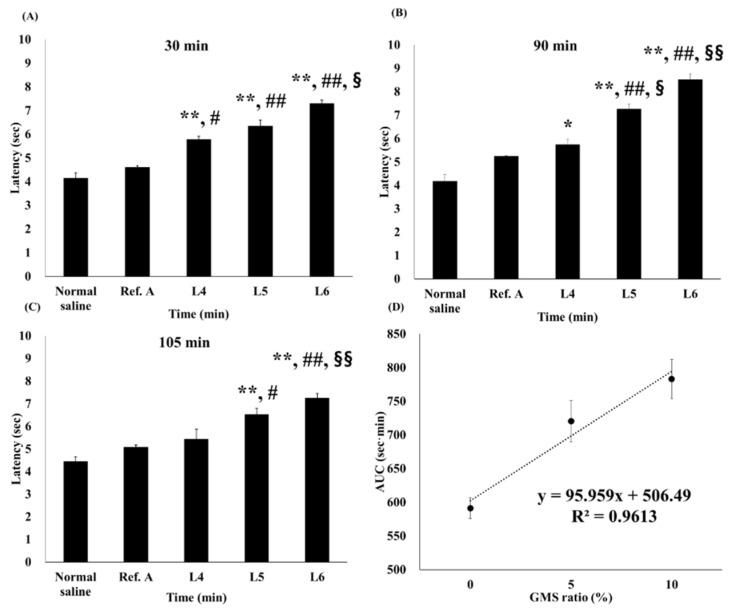
In vivo tail flick latency vs. time passed; comparison by concentration. The latency of in vivo tail-flick test at (**A**) 30, (**B**) 90 and, (**C**) 105 min; (**D**) Correlation between area under the curve (AUC) and glycerol monostearate (GMS) ratio (%) (mean ± standard error, *n* = 4). * ANOVA, LSD, *p* < 0.05, compared with the normal saline group. # ANOVA, LSD, *p*-value < 0.05, compared with Ref. A group. § ANOVA, LSD, *p*-value < 0.05 compared with L4 group. ** ANOVA, LSD, *p*-value < 0.005 compared with Normal saline group. ## ANOVA, LSD, *p*-value < 0.005 compared with Ref. A group. §§ ANOVA, LSD, *p*-value < 0.005 compared with L4 group.

**Table 1 pharmaceutics-13-01468-t001:** Formulation and rheological characterization of lidocaine ointments.

Excipient	L1	L2	L3	L4	L5	L6
Petrolatum	95.00	91.00	86.00	91.95	86.45	81.45
MCT oil	1.00	1.00	1.00	1.00	1.00	1.00
Glycerol monostearate	-	5.00	10.00	-	5.00	10.00
Vitamin E-acetate	-	-	-	3.00	3.00	3.00
Lidocaine base	3.00	3.00	3.00	3.00	3.00	3.00
Allantoin	-	-	-	1.00	1.00	1.00
Prednisolone acetate	-	-	-	0.05	0.05	0.05
dl-Methylephedrine HCl	-	-	-	-	0.5	0.5
pH	8.4	8.4	8.5	8.5	8.5	8.5
Viscosity (×10^3^ cP)	-	-	-	23.5 ± 0.5	41.9 ± 0.8	95.2 ± 0.9
Minimum extrusion force (N)	-	-	-	19.5 ± 0.2	25.2 ± 0.8	54.9 ± 0.9

Abbreviations: MCT, Medium-chain triglyceride, HCl, hydrochloride.

**Table 2 pharmaceutics-13-01468-t002:** Droplet size of lidocaine ointment after dilution.

Ointment	Droplet Size (nm)	PDI
L1	1132.0 ± 118.8	0.22 ± 0.11
L2	117.0 ± 0.3	0.28 ± 0.01
L3	108.0 ± 1.4	0.30 ± 0.02

Abbreviation: PDI, polydispersity index.

**Table 3 pharmaceutics-13-01468-t003:** Area under the curve (AUC) of latency profile and effect ratio for the tail-flick tests.

	Normal Saline	Ref. A	L4	L5	L6
AUC (sec × min)	439.8 ± 32.6	512.3 ± 7.2 *	583.0 ± 14.3 **^, #^	706.1 ± 24.9 **^, ##, §§^	778.9 ± 26.3 **^, ##, §§, †^
Effect fold compared to Normal saline	1.00	1.16	1.33	1.61	1.77
Effect fold compared to Ref. A	0.86	1.00	1.14	1.38	1.52

* ANOVA, LSD, *p*-value < 0.05, compared with normal saline group. ^#^ ANOVA, LSD, *p*-value < 0.05, compared with Ref. A group. ^†^ ANOVA, LSD, *p*-value < 0.05 compared with L5 group. ** ANOVA, LSD, *p*-value < 0.005 compared with normal saline group. ^##^ ANOVA, LSD, *p*-value < 0.005 compared with Ref. A group. ^§§^ ANOVA, LSD, *p*-value < 0.005 compared with L4 group.

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
