# Peer review of "Preparation and In Vivo Evaluation of a Lidocaine Self-Nanoemulsifying Ointment with Glycerol Monostearate for Local Delivery"

_pharmaceutics, 2021, doi:10.3390/pharmaceutics13091468_

Round 1

Reviewer 1 Report

This manuscript reported the formulations of using glycerol monostearate as the nanoemulsifying agent to deliver lidocaine. A series of formulations with varying amounts of ingredients was studied. The materials were characterized by DSC, TEM and Raman spectroscopy. In vivo results demonstrated the best formulation with 5% of the nanoemulsifying agent.

In Table 2, the droplet size of ointment L1 is reported as 971+-387.5 nm. This is a large number (size) with a large error term. However, the PDI is only a small number 0.13. Are the data presented here accurate? More discussions should be included.

In Table 2, the PDI values of L2 and L3 should be changed to two decimal places, to be consistent.

Table 1, HCl instead of HCL.

Author Response

Reviewer Comments and Responses

Thank you for your thorough review and salient observations of this manuscript and for the comments and suggestions, which help to improve the quality of this manuscript. Our response follows.

Reviewer 1

This manuscript reported the formulations of using glycerol monostearate as the nanoemulsifying agent to deliver lidocaine. A series of formulations with varying amounts of ingredients was studied. The materials were characterized by DSC, TEM and Raman spectroscopy. In vivo results demonstrated the best formulation with 5% of the nanoemulsifying agent.

  1. In Table 2, the droplet size of ointment L1 is reported as 971+-387.5 nm. This is a large number (size) with a large error term. However, the PDI is only a small number 0.13. Are the data presented here accurate? More discussions should be included.

Response: Following the comments, we ran the experiment again. As a result, the PDI was also low. Although the PDI of each sample showed a low value and low deviation, the droplet size result showed a large deviation. This is considered to be insufficient reproducibility of droplet formation of L1 (Table 2)

  1. In Table 2, the PDI values of L2 and L3 should be changed to two decimal places, to be consistent.

Response: As recommended, we modified Table 2

  1. Table 1, HCl instead of HCL.

Response: As recommended, we modified Table 1

Reviewer 2 Report

In the manuscript entitled, Preparation and in vivo evaluation of a lidocaine self-nanoemulsifying ointment with glycerol monostearate for local delivery” the authors have examined the perspective of a self-emulsifying ointment containing lidocaine to treat hemorrhoids. The authors used an emulsifier that could produce nanoemulsions and provide skin deposition, which in turn can improve the drug diffusion. Various physicochemical properties, drug release, diffusion and in vivo effect were assessed. The data observed here signifies that the ointment with 5% emulsifier showed greater anesthetic effect. Overall, the presentation is good and will be interesting to the pharmaceutics readers. I have a few suggestions to improve the quality of the manuscript.

  1. Brief the advantages of the current study over other carrier systems of lidocaine in the literature.
  2. Authors may include the influence of other excipients in the formulation in drug transport?
  3. How about the drug-excipient interactions?
  4. Do the authors check the pH, viscosity, spreadability etc. of the formulations?
  5. Can authors discuss the stability of the formulations?

Author Response

Reviewer Comments and Responses

Thank you for your thorough review and salient observations of this manuscript and for the comments and suggestions, which help to improve the quality of this manuscript. Our response follows.

Reviewer 2

In the manuscript entitled, Preparation and in vivo evaluation of a lidocaine self-nanoemulsifying ointment with glycerol monostearate for local delivery” the authors have examined the perspective of a self-emulsifying ointment containing lidocaine to treat hemorrhoids. The authors used an emulsifier that could produce nanoemulsions and provide skin deposition, which in turn can improve the drug diffusion. Various physicochemical properties, drug release, diffusion and in vivo effect were assessed. The data observed here signifies that the ointment with 5% emulsifier showed greater anesthetic effect. Overall, the presentation is good and will be interesting to the pharmaceutics readers. I have a few suggestions to improve the quality of the manuscript.

  1. Brief the advantages of the current study over other carrier systems of lidocaine in the literature.

    Response: We added “This study aimed to improve lidocaine ointments to relieve hemorrhoid pain with a petrolatum base [42, 43]. The petrolatum is effective in relieving itching from hemorrhoids. However, petrolatum base ointment is vulnerable to drug delivery due to slow drug release [44]. The self-emulsifying effect of GMS will be able to overcome the disadvantages of petrolatum base ointment.” On page 2-3

  2. Authors may include the influence of other excipients in the formulation in drug transport?

    Response: We added “It is thought that MCT oil forms a core as an oil phase, and GMS as a surfactant surrounds the surface to form nanosized droplets [52].” On page 14

  3. How about the drug-excipient interactions?

    Response: Unfortunately, stability test and compatibility test was not performed. Our aim was to evaluate nano-emulsifying effect of GMS, so we did not evaluate the stability aspects. We would like to thank you for your kind comments and will continue our work as a further study in the future.

  4. Do the authors check the pH, viscosity, spreadability etc. of the formulations?

    Response: We added rheological properties of ointment in Table 1 and that methods on page 3

  5. Can authors discuss the stability of the formulations?

    Response: We will continue our work as a further study in the future.

Reviewer 3 Report

Dear Authors,

I found this manuscript interesting, but I have several comments:

  1. Why were lidocaine ointments (L1, L2, L3) tested on rat skin and ointments (L4, L5, L6, and Ref. A) tested on human skin? Rat skin is more permeable than human skin and this may affect the results. 5 page
  2. What is the area of skin through which lidocaine penetrated? 5 page

Author Response

Reviewer Comments and Responses

Thank you for your thorough review and salient observations of this manuscript and for the comments and suggestions, which help to improve the quality of this manuscript. Our response follows.

Reviewer 3

Dear Authors,

I found this manuscript interesting, but I have several comments:

  1. Why were lidocaine ointments (L1, L2, L3) tested on rat skin and ointments (L4, L5, L6, and Ref. A) tested on human skin? Rat skin is more permeable than human skin and this may affect the results. 5 page

    Response: We used rat skin for initial L1, L2, L3 formulation studies because it was difficult to supply enough human cadaver skin.

  2. What is the area of skin through which lidocaine penetrated? 5 page

    Response: The ex vivo penetration test was conducted at about 2.0 cm2, and Raman mapping analyzed 100 μm x 100 μm. We added this information on page 5.
